# SAOT: Self-Supervised Continual Graph Learning with Structure-Aware Optimal Transport

Yuting Zhang [1]   Yanbei Liu [2]   Zhitao Xiao [2]   Lei Geng [2]   Yanwei Pang [3]   Xiao Wang [4]

## Abstract

Self-supervised Continual Graph Learning (CGL) aims to successively learn from a graph sequence with different tasks without label supervision—a paradigm that has attracted widespread attention. Most existing self-supervised CGL methods rely on instance-level consistency objectives that enforce stability of individual node (or node-pair) embeddings. Due to optimizing nodes in isolation, these methods fail to maintain global relational structure, causing inter-node correspondences to progressively distort under continual learning. To this end, we propose a novel Structure-Aware Optimal Transport (SAOT) framework that explicitly captures and preserves relational structure within graph representations across sequential tasks. Specifically, SAOT leverages optimal transport theory to capture global inter-node correspondences, thereby facilitating and enhancing graph representation learning. Simultaneously, SAOT incorporates a cross-task knowledge distillation mechanism to preserve the previous structural knowledge. Extensive experiments on four CGL benchmark datasets demonstrate that SAOT outperforms existing self-supervised baselines. In particular, SAOT achieves significant performance gains, improving average accuracy by up to 5% on CoraFull-CL and over 15% on Products-CL compared with state-of-the-art methods in the Class-IL setting.

## 1. Introduction

Graph data are ubiquitous in real-world applications, modeling complex systems with rich relational dependencies, such as citation networks, e-commerce systems, and biochemical molecules (Hamilton et al., 2017; Wu et al., 2021). Despite the remarkable progress of graph representation learning, most existing methods are designed for static settings, where the data distribution is assumed to be stationary (Kipf & Welling, 2017; Veličković et al., 2018). In contrast, real-world graph data are continuously generated, with new nodes, edges, or tasks appearing over time (Wang et al., 2020). For instance, papers on new research topics continuously enter citation networks, and novel molecular properties are progressively encountered in drug discovery tasks (Liu et al., 2021; Zhang et al., 2022a). To cope with such evolving data, graph models are required to incrementally acquire new knowledge while maintaining performance on previously learned tasks, a learning paradigm known as Continual Graph Learning (CGL) (Zhou & Cao, 2021; Zhang et al., 2022a). However, simply training models sequentially on incoming data is prone to catastrophic forgetting (McCloskey & Cohen, 1989; Goodfellow et al., 2014), while retraining a model on all accumulated data is computationally expensive and often infeasible when historical data are unavailable.

To mitigate catastrophic forgetting, existing CGL approaches generally are divided into three categories. Parameter isolation methods (Zhang et al., 2023; Cai et al., 2022) allocate task-specific parameters or modules to avoid interference between tasks. Regularization-based approaches (Kirkpatrick et al., 2017; Liu et al., 2021) constrain parameter updates by penalizing changes to important parameters or outputs learned from previous tasks. Replay-based methods (Zhou & Cao, 2021; Zhang et al., 2022b) explicitly store a small set of historical nodes or subgraphs and rehearse them during training to maintain past knowledge. While these approaches have demonstrated effectiveness, they typically rely on explicit supervision. However, annotating complex graph data is often time-consuming, labor-intensive, and sometimes even impractical, especially when graphs continuously emerge in an online manner.

Consequently, it is crucial to advance self-supervised con-

[1]School of Electronics and Information Engineering, Tiangong University, Tianjin, China [2]School of Life Sciences, Tiangong University, Tianjin, China [3]School of Electrical and Infomation Engineering, Tianjin University, Tianjin, China [4]School of Software, Beihang University, Beijing, China. Correspondence to: Yanbei Liu <liuyanbei@tiangong.edu>.

*Proceedings of the 43rd International Conference on Machine Learning*, Seoul, South Korea. PMLR 306, 2026. Copyright 2026 by the author(s).

tinual graph learning, enabling models to acquire knowledge directly from unlabeled streaming data. Driven by the above necessity, recent studies (Sun et al., 2023; Peng et al., 2025) have explored self-supervised approaches for continual graph learning scenarios. In particular, TRACE (Peng et al., 2025) provides a systematic empirical study of representative self-supervised graph representation learning paradigms in continual settings, demonstrating that self-supervised graph models (Kipf & Welling, 2016; You et al., 2020b; Bielak et al., 2022) can learn more transferable and stable representations than their supervised counterparts in the absence of label supervision.

However, most existing self-supervised CGL methods rely on instance-level consistency objectives, treating each node (or node pair) as an independent learning instance to stabilize individual node embeddings (Sun et al., 2023; Peng et al., 2025). Although these methods effectively promote local embedding stability, they overlook a critical characteristic of streaming graph data: long-term knowledge is encoded not merely in isolated embeddings but in the relational structure within the representation space. Consequently, even if individual nodes embeddings remain locally stable, the relationships among nodes can gradually distort—a phenomenon known as structural drift, which manifests as changes in relative distances or the overall organization of node clusters across sequential tasks. Because most existing objectives do not explicitly model relational dependencies, such drift accumulates over time, ultimately impairing long-term knowledge retention.

To this end, we propose a novel self-supervised continual graph learning framework based on Structure-Aware Optimal Transport, named SAOT. Specifically, SAOT leverages Optimal Transport (OT) theory to encode the relational structure among all nodes. Within each task, SAOT constructs an optimal transport plan that jointly captures correspondences among nodes in the graph space. The plan is used as a structural reference to guide the encoder to learn structure-aware, transferable node representations. To mitigate structural drift caused by sequential tasks, SAOT employs cross-task knowledge distillation to preserve the previous structural information. Through the coordinated operation of the above two modules, SAOT enables the model to adapt effectively to new tasks while consolidating previously learned structural knowledge, achieving an optimal balance between plasticity and stability. In summary, the main contributions of the proposed SAOT are as follows:

- We propose SAOT, a novel self-supervised continual graph learning framework that leverages optimal transport to model and preserve relational structure in graph representations across sequential tasks.

- We design a cross-task knowledge distillation mechanism that preserves relational structure by distilling

optimal transport plans defined in the representation space across tasks, thereby effectively mitigating the problem of structure drift under continual learning.

- We conduct extensive experiments on four CGL benchmark datasets under both Class-IL and Task-IL settings. Experimental results demonstrate that the proposed SAOT outperforms existing self-supervised baselines. In particular, in the Class-IL setting, SAOT achieves an improvement of up to 5% on CoraFull-CL and over 15% on Products-CL in terms of average accuracy compared with state-of-the-art methods.

## 2. Related Work

### 2.1. Continual Graph Learning

Inspired by recent advances in continual learning for computer vision, various methods for CGL have been proposed in recent years to address learning on streaming graph data (Babakniya et al., 2024; Lee et al., 2024). From a methodological perspective, existing CGL methods can be broadly grouped into three categories. Parameter isolation methods (Cai et al., 2022; Zhang et al., 2023) allocate task-specific parameters or network components through specialized architectures, such as expanding networks, task-specific modules, or gated routing mechanisms, in order to prevent interference between tasks. Replay-based methods (Zhou & Cao, 2021; Zhang et al., 2022b; Liu et al., 2023; Zhang et al., 2024) mitigate catastrophic forgetting by maintaining a memory buffer that stores information from previous tasks and replaying it during the learning of new tasks. Regularization-based methods (Kirkpatrick et al., 2017; Liu et al., 2021; Sun et al., 2023) restrict parameter updates or representation changes across tasks by imposing consistency constraints on model parameters, node embeddings, or output predictions.

However, most existing CGL methods rely heavily on supervisory signals, such as node or graph labels, which limits their applicability in realistic scenarios. In contrast, research on unsupervised or self-supervised CGL remains relatively underexplored. More recently, a notable attempt is RieGrace (Sun et al., 2023), which proposes a unified framework that combines GNNs with CurvNet. It explicitly handles task-specific adaptation in Riemannian space and employs a label-free Lorentz distillation mechanism to mitigate catastrophic forgetting. Inspired by the Complementary Learning Systems theory, TRACE (Peng et al., 2025) employs a dual-system framework to mitigate catastrophic forgetting without external supervision. Specifically, a fast-learning system extracts key knowledge through node proxies, while a slow-learning system consolidates memory via adaptive spaced replay.

## 2.2. Self-Supervised Graph Learning

Self-supervised graph learning aims to learn expressive node or graph representations without relying on explicit labels by exploiting intrinsic structural and attribute-based signals. Early approaches (Kipf & Welling, 2016; Jin et al., 2020) are primarily based on generative objectives, such as graph autoencoders and variational graph autoencoders, which reconstruct node features or graph structures from latent representations. More recent methods (Veličković et al., 2019; You et al., 2020a; Zhu et al., 2020) adopt contrastive learning paradigms, where representations are learned by maximizing agreement between different augmented views of the same graph while contrasting them against other samples. These approaches have demonstrated good performance on various downstream tasks. Beyond contrastive learning, redundancy-reduction and non-contrastive paradigms (Lee et al., 2022; Thakoor et al., 2022; Bielak et al., 2022) have been proposed to remove the reliance on negative samples. These methods learn invariant representations across augmented views by explicitly minimizing feature redundancy or aligning cross-view statistics. Nevertheless, most methods are trained offline under the assumption of full data availability, which does not reflect the reality that graph data arrive sequentially over time.

# 3. Preliminaries

In this section, we briefly review optimal transport, which serve as the key technical foundation of our method. We then formally defined the studied paradigm: self-supervised continual graph learning.

## 3.1. Plan and Optimal Transport

The optimal transport problem was originally proposed to study the most cost-effective way of transporting the shape of one pile of sand into the shape of another (Wang et al., 2023). Specifically, it studies how to transform distribution $\mu$ into another distribution $\nu$ with the minimum total transportation cost (i.e, the optimal transportation distance). This transport plan is called the optimal plan $\pi$, where the element of $\pi$ describes the probability of moving mass from one position to another. Formally, given two sets of features $\mathbf{X}_1 = \{\mathbf{X}_1^i\}_{i=1}^n$ and $\mathbf{X}_2 = \{\mathbf{X}_2^j\}_{j=1}^m$, where $n$ and $m$ are the number of features, $\mu \in \mathbb{R}^n$ and $\nu \in \mathbb{R}^m$ are the probability distributions of the entities in the two sets, respectively. The formulation of the OT distance is

$$\mathcal{D}(\mathbf{X}_1, \mathbf{X}_2) = \min_{\pi \in \Pi(\mu,\nu)} \sum_{i \in \llbracket n \rrbracket} \sum_{j \in \llbracket m \rrbracket} c_{\mathcal{X}}(\mathbf{X}_1^i, \mathbf{X}_2^j) \cdot \pi_{ij}, \quad (1)$$

abbreviated as

$$\mathcal{D}(\mathbf{X}_1, \mathbf{X}_2) = \min_{\pi \in \Pi(\mu,\nu)} \langle \mathcal{K}(\mathbf{X}_1, \mathbf{X}_2), \pi \rangle, \quad (2)$$

where $\pi \in \Pi(\mu, \nu) = \{\pi \in \mathbb{R}^{n \times m} \mid \pi \mathbf{1}_m = \mu, \pi^\top \mathbf{1}_n = \nu\}$ denotes all the joint distributions $\pi$ with the marginal distribution $\mu$ and $\nu$. $\mathcal{K}(\mathbf{X}_1, \mathbf{X}_2)_{ij} = c_{\mathcal{X}}(\mathbf{X}_1^i, \mathbf{X}_2^j)$ is the cost of moving $\mathbf{X}_1^i$ to $\mathbf{X}_2^j$. The cost can be calculated by cosine distance. The $\mathbf{1}$ denotes a vector that elements are all 1, $\llbracket n \rrbracket = \{1, 2, \ldots, n\}$, and $\langle \cdot, \cdot \rangle$ is the inner product operator. The $\pi \in \mathbb{R}^{n \times m}$ is called as transport plan. The distance $\mathcal{D}(\cdot, \cdot)$ is also known as Wasserstein distance. In this work, we focus on the case of discrete distributions on graphs.

## 3.2. Problem Formulation

The setting of continual graph learning assumes a sequence of disjoint tasks $\mathcal{T} = \{\mathcal{T}_1, \ldots, \mathcal{T}_T\}$ which will be encountered sequentially. Each task $\mathcal{T}_t \in \mathcal{T}$ is defined on a graph $\mathcal{G}_t = (\mathcal{V}_t, \mathcal{E}_t)$, where $\mathcal{V}_t$ and $\mathcal{E}_t$ denote the sets of nodes and edges, respectively. The node attributes are represented by the feature matrix $\mathbf{X}_t \in \mathbb{R}^{|\mathcal{V}_t| \times d}$, where $d$ is the feature dimension. Furthermore, the topological structure of the graph is captured by the adjacency matrix $\mathbf{A}_t \in \{0, 1\}^{|\mathcal{V}_t| \times |\mathcal{V}_t|}$. In this work, we follow a decoupled representation–classifier paradigm (Cha et al., 2021; Gomez-Villa et al., 2022; Sun et al., 2023) for self-supervised CGL, in which representation learning is explicitly separated from downstream classifier training.

Given a sequence of graph related tasks $\mathcal{T}$, we aim at learning a shared graph encoder $\mathcal{F}_\xi : \mathcal{V}_t \to \mathbb{R}^H$ across all task sequences, which maps nodes to $H$-dimensional representation vectors, so that the encoder is able to continuously adapt new knowledge in the current tasks without catastrophically forgetting knowledge acquired from previous tasks. During the learning process, no external supervisory signals (i.e., labels) are involved.

To evaluate the quality of learned representations, we follow the experimental setting in prior work on self-supervised continual learning (Sun et al., 2023; Gomez-Villa et al., 2024; Peng et al., 2025). Specifically, for the graph $\mathcal{G}_t$ from each task, we have training node set $\mathcal{V}_t^{tr}$ and testing node set $\mathcal{V}_t^{te}$. Each node $v_t^i$ in $\mathcal{G}_t$ belongs to a category $y_t^i \in \mathcal{Y}_P$, where $\mathcal{Y}_P$ denotes the label set of $P$ categories involved in all tasks. During evaluation, the parameters of the graph encoder $\mathcal{F}_\xi$ are kept frozen, and we provide the learned embeddings across the training node set to the linear classifier for performance evaluation. We consider two different settings, i.e., Task Incremental Learning (Task-IL) and Class Incremental Learning (Class-IL). In the Task-IL setting, the model is only required to distinguish different classes within each task without considering the classes that have already appeared from existing tasks. In the Class-IL setting, the model is required to distinguish among all classes from the current and previous tasks. Obviously, Class-IL is more realistic and challenging.

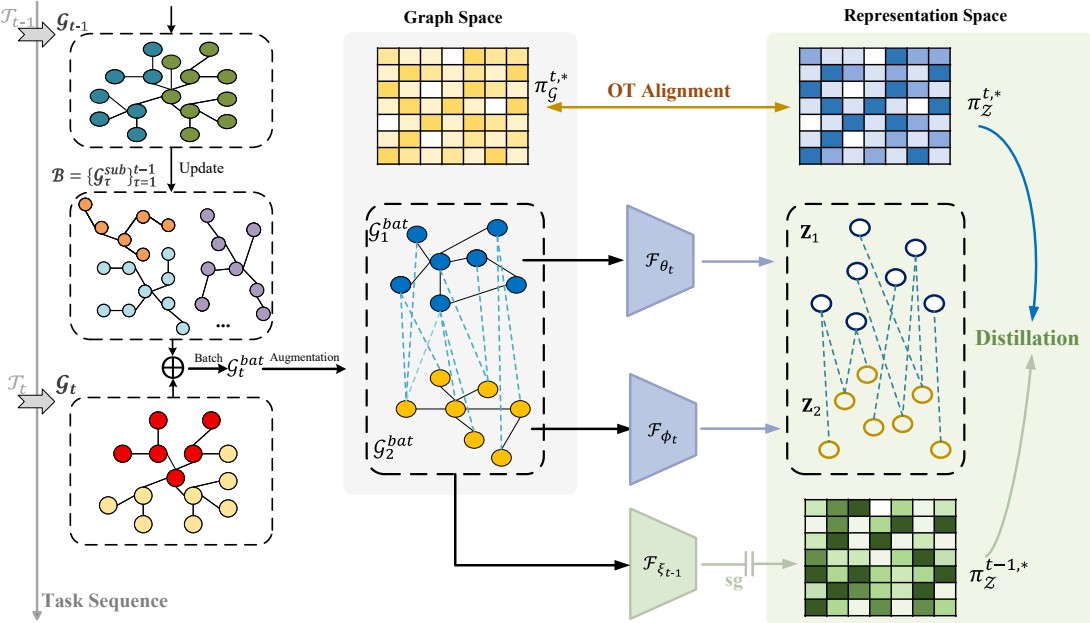

*Figure 1.* **Overview of SAOT.** It leverages optimal transport mechanism to learn and preserve relational structure in CGL. When a new task arrives, the model learns structure-aware representations by aligning optimal transport plans in the graph and embedding spaces. Simultaneously, SAOT consolidates the learned structural knowledge from the previous model by distillation mechanism.

## 4. Methodology

This section describes the details of the proposed SAOT framework. SAOT consists of two components: (i) Graph Representation via Optimal Transport, which learns structure-aware representations within each task, and (ii) Cross-task Knowledge Distillation, which preserves relational knowledge across sequential tasks. The framework details are illustrated in Figure 1.

### 4.1. Graph Representation via Optimal Transport

The primary objective of our framework is to maximize plasticity on new tasks. Specifically, we expect to fully adapt the current input graph for acquiring high-quality knowledge. To this end, we design a replay-based graph representation learning framework based on optimal transport mechanism, which explicitly aligns relational structures between the input graph and the learned representation space, enabling structure-aware graph embeddings.

**Graph Augmentation.** For task $\mathcal{T}_t$, the training input data consists of the graph $\mathcal{G}_t = (\mathcal{V}_t, \mathcal{E}_t)$ and a set of historical subgraphs $\{\mathcal{G}_\tau^{sub}\}_{\tau=1}^{t-1}$ retrieved from the fixed-size memory buffer $\mathcal{B}$, where $\tau \in \{1, \ldots, t-1\}$ indexes the prior tasks. Each historical subgraph $\mathcal{G}_\tau^{sub}$ is constructed via layer-wise neighbor sampling to a fixed budget, ensuring bounded memory usage while preserving local structural information. During training, the current graph and replayed subgraphs

are consolidated into batched graph $\mathcal{G}_t^{bat}$ via the disjoint union operation. For simplicity, we omit the task index in the following and denote the batched graph as $\mathcal{G}^{bat}$. For the batched graph $\mathcal{G}^{bat}$, we construct two augmented views, $\mathcal{G}_1^{bat}$ and $\mathcal{G}_2^{bat}$, via a stochastic augmentation function that randomly drops edges and masks node features. Formally, we define two views as $\mathcal{G}_1^{bat} = (\mathbf{A}_1, \mathbf{X}_1, \mu)$ and $\mathcal{G}_2^{bat} = (\mathbf{A}_2, \mathbf{X}_2, \nu)$, where $\mu$ and $\nu$ are the empirical distributions of nodes in $\mathcal{G}_1^{bat}$ and $\mathcal{G}_2^{bat}$, respectively. $\mathbf{X}_1 \in \mathbb{R}^{n \times d}$ and $\mathbf{X}_2 \in \mathbb{R}^{m \times d}$ denote their feature matrices. $\mathbf{A}_1 \in \mathbb{R}^{n \times n}$ and $\mathbf{A}_2 \in \mathbb{R}^{m \times m}$ are adjacency matrices.

The underlying graph topology provides a valuable source of structure-level supervision for representation learning. However, directly leveraging the structural signal is non-trivial. Since node embeddings and graph topology reside in different metric spaces, it is challenging to establish comparable similarity metrics across these spaces. Optimal transport theory offers a solution for such comparison via the transport plan. In this work, we aim to find the optimal transport plan for two graphs by minimizing the transport cost (i.e.,optimal transport distance).

The Wasserstein distance (Equation 1) requires the two distributions to reside in the same space. But for graphs, it is difficult to measure the cost between two nodes on different graphs without node label (attribute). Even if there is some way to get the cost between nodes, the Wasserstein distance cannot consider the edge information. Therefore, we need

to take the edge information into account as follows

$$\mathcal{D}_{\text{GW}}(\mathbf{A}_1, \mathbf{A}_2) = \min_{\pi_{\mathcal{G}} \in \Pi(\mu, \nu)} \sum_{i,k \in [\![n]\!]^2} \sum_{j,l \in [\![m]\!]^2} \tag{3}$$
$$c_{\mathcal{A}}(\mathbf{A}_1^{ik}, \mathbf{A}_2^{jl}) \cdot \pi_{ik} \pi_{jl},$$

abbreviated as

$$\mathcal{D}_{\text{GW}}(\mathbf{A}_1, \mathbf{A}_2) = \min_{\pi_{\mathcal{G}} \in \Pi(\mu, \nu)} \langle \mathcal{L}(\mathbf{A}_1, \mathbf{A}_2) \otimes \pi, \pi \rangle, \tag{4}$$

where $\mathcal{L}(\mathbf{A}_1, \mathbf{A}_2)$ is 4-D tensor and $\mathcal{L}(\mathbf{A}_1, \mathbf{A}_2)_{ijkl} = c_{\mathcal{A}}(\mathbf{A}_1^{ik}, \mathbf{A}_2^{jl})$. The cost function $c_{\mathcal{A}}$ is defined as $c_{\mathcal{A}}(\mathbf{A}_1^{ik}, \mathbf{A}_2^{jl}) = |\mathbf{A}_1^{ik} - \mathbf{A}_2^{jl}|$, where $|\cdot|$ denotes the absolute value operator. $[\![n]\!]^2 = [\![n]\!] \times [\![n]\!]$, $\otimes$ denotes the tensor-matrix multiplication. The distance $\mathcal{D}_{\text{GW}}(\cdot, \cdot)$ is known as the Gromov-Wasserstein distance. Then we consider the complete graph including edge structure and node features. The fused Gromov–Wasserstein distance between graphs $\mathcal{G}_1^{bat}$ and $\mathcal{G}_2^{bat}$ can be defined as

$$\mathcal{D}_{\text{FGW}}(\mathcal{G}_1^{bat}, \mathcal{G}_2^{bat}) = \min_{\pi_{\mathcal{G}} \in \Pi(\mu, \nu)} \sigma \sum_{ij} c_{\mathcal{X}}(\mathbf{X}_1^i, \mathbf{X}_2^j) \cdot \pi_{ij}^{\mathcal{G}}$$
$$+ (1 - \sigma) \sum_{ijkl} c_{\mathcal{A}}(\mathbf{A}_1^{ik}, \mathbf{A}_2^{jl}) \cdot \pi_{ij}^{\mathcal{G}} \pi_{kl}^{\mathcal{G}}, \tag{5}$$

which is equivalent to

$$\min_{\pi_{\mathcal{G}} \in \Pi(\mu, \nu)} \langle \sigma \mathcal{K}(\mathbf{X}_1, \mathbf{X}_2) + (1 - \sigma) \mathcal{L}(\mathbf{A}_1, \mathbf{A}_2) \otimes \pi_{\mathcal{G}}, \pi_{\mathcal{G}} \rangle, \tag{6}$$

where $\sigma \in [0; 1]$ represents the trade-off parameter for adjusting nodes and edges.

**Optimal Transport Plan.** In this work, we aim to learn structure-aware representations while preserving nodel-level semantic information. To this end, we integrate the node features and edge structure on the graph and design a new objective function to align the optimal transport plans derived from the graph space and the representation space. Specifically, the fused optimal transport plan $\pi_{\mathcal{G}}^* \in \mathbb{R}^{n \times m}$ between the two graphs $\mathcal{G}_1^{bat}$ and $\mathcal{G}_2^{bat}$ above can be defined by minimizing the fused Gromov-Wasserstein distance objective:

$$\pi_{\mathcal{G}}^* = \arg\min_{\pi_{\mathcal{G}} \in \Pi(\mu, \nu)} \langle \sigma \mathcal{K}(\mathbf{X}_1, \mathbf{X}_2) + (1 - \sigma) \mathcal{L}(\mathbf{A}_1, \mathbf{A}_2) \otimes \pi_{\mathcal{G}}, \pi_{\mathcal{G}} \rangle. \tag{7}$$

By tuning the parameter $\sigma$, we can control the deviation of the learned optimal transport plan between node features and edge structure.

Then we use the backbone model (e.g., GNNs) to obtain the node representations of graph. We can use Equation (1) directly to calculate the optimal plan $\pi_{\mathcal{Z}}^* \in \mathbb{R}^{n \times m}$ as

$$\pi_{\mathcal{Z}}^* = \arg\min_{\pi_{\mathcal{Z}} \in \Pi(\mu, \nu)} \langle \mathcal{R}(\mathbf{Z}_1, \mathbf{Z}_2), \pi_{\mathcal{Z}} \rangle, \tag{8}$$

where $\mathcal{R}(\mathbf{Z}_1, \mathbf{Z}_2)_{ij} = c_{\mathcal{Z}}(\mathbf{Z}_1^i, \mathbf{Z}_2^j)$, $\mathbf{Z}_1$ and $\mathbf{Z}_2$ denote the node representations corresponding to $\mathcal{G}_1^{bat}$ and $\mathcal{G}_2^{bat}$, respectively.

**Optimal Transport Alignment.** Obviously, if the node embeddings learned by the encoder can preserve the relational correspondences observed in the graph space, they are likely to yield more informative representations. Therefore, we force the encoder to preserve matching relationships in graph space by aligning the optimal transport plan defined on the input graphs with that computed from their node embeddings. We minimize the discrepancy between the optimal transport plans in the two spaces as the alignment loss as follows

$$\mathcal{L}_{mat} = \Theta(\boldsymbol{\pi}_{\mathcal{G}}^*, \boldsymbol{\pi}_{\mathcal{Z}}^*), \tag{9}$$

where $\Theta(\cdot, \cdot)$ denotes a discrepancy measure, such as the KL divergence or the Frobenius norm.

Additionally, to guide the encoder to learn a representation retaining the structural information of the input graph, we calibrate the cost matrix $\mathcal{R}(\mathbf{Z}_1, \mathbf{Z}_2)$, which captures the implicit relational structure between nodes in the representation space, as follows

$$\mathcal{L}_{str} = \Theta\Big(\sigma \mathcal{K}(\mathbf{X}_1, \mathbf{X}_2) + (1 - \sigma) \mathcal{L}(\mathbf{A}_1, \mathbf{A}_2) \otimes \pi_{\mathcal{G}}^*, \mathcal{R}(\mathbf{Z}_1, \mathbf{Z}_2)\Big). \tag{10}$$

As a result, the overall intra-task graph transport alignment loss is defined as

$$\mathcal{L}_{gta} = \mathcal{L}_{mat} + \alpha \mathcal{L}_{str}, \tag{11}$$

where $\alpha$ is the trade-off parameter. The alignment loss encourages the encoder to learn the relational structure of the input graph in the learned node embeddings.

## 4.2. Cross-task Knowledge Distillation

Continuous optimization on new tasks inevitably induces representation drift, causing the structural knowledge (i.e., relational dependencies) acquired from prior tasks to deteriorate. To consolidate past and present knowledge, we design a cross-task knowledge distillation mechanism, regulating the changes of relational structure. Formally, let $\xi_{t-1}$ denote the parameters of the encoder after completing task $\mathcal{T}_{t-1}$, which is then frozen to serve as a reference teacher model $\mathcal{F}_{\xi_{t-1}}$ for the current task $\mathcal{T}_t$. Given the two augmented graphs $\mathcal{G}_1^{bat}$ and $\mathcal{G}_2^{bat}$, we compute a reference optimal transport plan $\pi_{\mathcal{Z}}^{t-1,*}$ using the teacher model $\mathcal{F}_{\xi_{t-1}}$. The cross-task structural knowledge distillation loss is defined as

$$\mathcal{L}_{skd} = \Theta\left(\pi_{\mathcal{Z}}^{t-1,*}, \pi_{\mathcal{Z}}^{t,*}\right), \tag{12}$$

*Table 1.* Statistics of datasets and task splittings.

| Dataset | CoraFull-CL | Arxiv-CL | Reddit-CL | Products-CL |
|---|---|---|---|---|
| # Nodes | 19,793 | 169,343 | 232,965 | 2,449,029 |
| # Edges | 130,622 | 1,166,243 | 114,615,892 | 61,859,140 |
| # Classes | 70 | 40 | 40 | 47 |
| # Tasks | 35 | 20 | 20 | 23 |

where $\Theta(\cdot, \cdot)$ denotes the divergence measure between optimal transport plans.

To this end, the overall loss function is composed of graph transport alignment loss $\mathcal{L}_{gta}$ and distillation loss $\mathcal{L}_{skd}$ as follows

$$\mathcal{L}_{total} = \mathcal{L}_{gta} + \beta\mathcal{L}_{skd} = \mathcal{L}_{mat} + \alpha\mathcal{L}_{str} + \beta\mathcal{L}_{skd}, \quad (13)$$

where $\beta$ controls the strength of cross-task structural regularization. Under the guidance of the overall objective function, SAOT integrates newly acquired representations with previously learned structural knowledge in the representation space, and progressively consolidates them into the model.

## 5. Experiments

### 5.1. Datasets

Following the Continual Graph Learning Benchmark (CGLB) (Zhang et al., 2022a), we conduct experiments on four public benchmark datasets: CoraFull-CL, Arxiv-CL, Reddit-CL, and Products-CL. For each dataset, node classes are split into a sequence of disjoint tasks with a fixed order, where each task contains two classes, and the original graph is accordingly split into task-specific subgraphs without inter-task edges. In the continual learning setting, an increasing number of tasks leads to more severe forgetting. The setting with the maximum number of tasks allows us to thoroughly assess the limitations of different CGL methods. Detailed dataset statistics and task splittings are summarized in Table 1.

### 5.2. Baselines

To demonstrate the effectiveness of SAOT, we compare with representative baselines from two distinct categories. **Self-supervised CGL Methods:** RieGrace (Sun et al., 2023) and TRACE (Peng et al., 2025) are standard self-supervised CGL models. In addition, we adapt representative self-supervised graph learning models, i.e., GAE (Kipf & Welling, 2016), DGI (Veličković et al., 2019), Graph Barlow Twins (G-BT) (Bielak et al., 2022), and various GCN contrastive variants (GCN-Clu, GCN-Par, GCN-Comp) (You et al., 2020b) to the continual learning setting by training on each task sequentially. **Supervised CGL Methods:** For a comprehensive comparison, we include supervised approaches, i.e, TWP (Liu et al., 2021), ER-GNN (Zhou &

Cao, 2021), CaT (Liu et al., 2023), TACO (Han et al., 2024), and PUMA (Liu et al., 2025). Additionally, we perform the joint training strategy, where the model is trained on all tasks simultaneously without replay, serving as a empirical upper bound for continual learning.

### 5.3. Metrics and Parameter Settings

To comprehensively evaluate the performance of CGL models, we follow standard evaluation metrics widely adopted in prior works (Jin et al., 2020; Zhou & Cao, 2021). We construct a performance matrix $\mathbf{M}^p \in \mathbb{R}^{T \times T}$, where $M_{i,j}^p$ denotes the classification accuracy on task $j$ after the model has been trained on the first $i$ tasks. Based on $\mathbf{M}^p$, we adopt two commonly used metrics: Average Performance (AP, also called average accuracy) and Average Forgetting (AF, also called backward transfer). AP measures the overall predictive performance after completing all tasks and is defined as $\text{AP} = \frac{1}{T}\sum_{i=1}^{T} M_{T,i}^p$. AF quantifies the degree of catastrophic forgetting across tasks and is computed as $\text{AF} = \frac{1}{T-1}\sum_{i=1}^{T-1}\left(M_{T,i}^p - M_{i,i}^p\right)$. A negative AF indicates severe forgetting of previously learned tasks, whereas a positive AF suggests effective knowledge retention. When models achieve similar AP, a higher AF is preferable.

**Implementation Details.** All experiments are implemented in PyTorch and conducted on NVIDIA L40S GPUs with 48GB memory. For baseline methods, we follow the experimental settings reported in their original papers. For the proposed SAOT, we employ a 2-layer GCN encoder with a hidden dimension of 512 on Arxiv-CL, CoraFull-CL, and Products-CL. We switch to a 3-layer GAT encoder with four attention heads on Reddit-CL due to its much higher edge density and extremely large neighborhoods. The replay buffer size is set to 800 samples per task for Arxiv-CL, Products-CL, and Reddit-CL, while a smaller budget of 200 samples per task is used for CoraFull-CL. The learning rate is tuned via a "grid-search" strategy ranging from 1e-4 to 1e-2, and all models are optimized by using Adam (Kingma & Ba, 2015). All results are averaged over five independent runs with different random seeds, and we report the mean and standard deviation.

### 5.4. Experimental Results

Table 2 and Table 3 summarize the overall performance of all methods under Class-IL and Task-IL settings. The best results among self-supervised methods are highlighted in bold, and the second-best are underlined. Across all datasets, SAOT consistently achieves the best AP performance while effectively mitigating catastrophic forgetting.

**Class-IL Scenario.** As shown in Table 2, under the challenging Class-IL scenario, SAOT outperforms existing self-

*Table 2.* Performance comparison under the Class-IL setting without inter-task edges.

| Type | Method | CoraFull-CL | | Arxiv-CL | | Reddit-CL | | Products-CL | |
|---|---|---|---|---|---|---|---|---|---|
| | | AP/% | AF/% | AP/% | AF/% | AP/% | AF/% | AP/% | AF/% |
| Full | Joint | 71.4±0.3 | - | 51.9±0.4 | - | 91.7±0.2 | - | 15.7±0.1 | - |
| Supervised | TWP | 20.9±3.8 | -73.3±4.1 | 4.9±0.0 | -89.0±0.4 | 13.5±2.6 | -89.7±2.7 | 3.0±0.7 | -89.7±1.0 |
| | ER-GNN | 3.0±0.1 | -93.8±0.5 | 30.3±1.5 | -54.0±1.3 | 88.5±2.3 | -10.8±2.4 | 24.5±1.9 | -67.4±1.9 |
| | CaT | 68.5±0.9 | -5.7±1.3 | 64.9±0.3 | -12.5±0.8 | 97.7±0.1 | -0.4±0.1 | 71.1±0.3 | -5.4±0.3 |
| | TACO | 54.3±1.0 | -15.6±2.1 | 25.7±0.7 | -19.4±1.7 | 83.7±0.4 | -8.6±0.4 | 11.3±0.5 | -6.6±0.7 |
| | PUMA | 77.9±0.2 | -4.2±0.9 | 67.0±0.1 | -12.2±0.3 | 98.0±0.1 | -0.3±0.1 | 74.2±0.4 | -4.1±0.5 |
| Self-Supervised | GAE | 58.1±0.2 | -3.1±0.4 | 29.7±0.2 | -21.7±0.1 | 90.6±0.3 | -2.3±0.2 | 5.9±0.2 | -7.0±0.1 |
| | DGI | 6.5±0.4 | -23.3±0.5 | 22.0±0.2 | -22.4±0.1 | 56.8±0.3 | -10.9±0.2 | 4.3±0.1 | -5.3±0.2 |
| | G-BT | 57.7±0.4 | -0.3±0.8 | 44.5±1.5 | -17.9±1.0 | 96.2±0.1 | -1.1±0.6 | 25.4±1.6 | -14.7±1.4 |
| | GCN-Clu | 4.6±0.1 | -49.9±0.4 | 25.4±0.5 | -22.3±0.3 | 84.1±0.2 | -6.7±0.1 | 7.8±0.2 | -8.4±0.6 |
| | GCN-Par | 10.4±0.1 | -44.1±0.1 | 41.5±0.4 | -24.5±0.6 | 92.8±0.3 | -2.5±0.3 | 7.8±0.1 | -14.2±0.2 |
| | GCN-Comp | 6.6±0.3 | -35.1±0.2 | 9.1±0.1 | -28.0±0.2 | 87.5±0.1 | -3.3±0.2 | 6.1±0.3 | -5.3±0.2 |
| | RieGrace | 3.3±0.1 | -11.4±0.3 | 4.9±0.1 | -18.0±0.1 | 4.3±0.1 | -8.5±0.1 | 4.1±0.1 | -4.9±0.1 |
| | TRACE | 71.2±0.2 | -6.2±0.5 | 47.8±0.1 | **-6.4±0.2** | 98.1±0.1 | -0.2±0.1 | 25.6±0.3 | **-0.3±0.2** |
| | **SAOT** | **76.3±0.3** | **-0.2±0.6** | **51.6±0.2** | -17.5±0.4 | **98.5±0.2** | **-0.2±0.1** | **40.7±0.3** | -0.6±0.2 |

*Table 3.* Performance comparison under the Task-IL setting without inter-task edges.

| Type | Method | CoraFull-CL | | Arxiv-CL | | Reddit-CL | | Products-CL | |
|---|---|---|---|---|---|---|---|---|---|
| | | AP/% | AF/% | AP/% | AF/% | AP/% | AF/% | AP/% | AF/% |
| Full | Joint | 94.5±0.1 | - | 92.6±0.2 | - | 98.5±0.1 | - | 85.9±0.1 | - |
| Supervised | TWP | 92.2±0.5 | -0.9±0.3 | 86.0±0.8 | -2.8±0.8 | 87.4±3.8 | -12.6±4.0 | 90.3±0.1 | -0.5±0.1 |
| | ER-GNN | 90.6±0.1 | -3.7±0.1 | 86.7±0.1 | 11.4±0.9 | 98.9±0.0 | -0.1±0.1 | 89.0±0.4 | -2.5±0.3 |
| | CaT | 93.3±0.4 | -0.3±0.6 | 94.7±0.3 | -0.8±0.3 | 99.3±0.0 | -0.0±0.1 | 94.9±0.3 | -0.5±0.5 |
| | TACO | 94.5±0.6 | -0.2±0.3 | 90.2±0.9 | 0.1±0.1 | 96.4±0.6 | -0.9±0.2 | 83.1±0.3 | -0.9±0.3 |
| | PUMA | 95.2±0.3 | -0.7±0.2 | 95.3±0.1 | 0.1±0.1 | 99.4±0.0 | 0.0±0.0 | 95.4±0.3 | 0.1±0.5 |
| Self-Supervised | GAE | 92.3±0.2 | 0.3±0.3 | 90.9±0.5 | -1.8±0.9 | 98.3±0.1 | -0.2±0.2 | 78.5±0.6 | -3.2±0.7 |
| | DGI | 91.5±0.2 | -2.4±0.5 | 90.1±0.2 | 0.2±0.1 | 93.7±0.1 | -0.2±0.1 | 76.4±0.1 | -0.3±0.2 |
| | G-BT | 94.2±0.3 | -0.4±0.2 | 93.0±0.4 | -0.2±0.2 | 99.3±0.1 | -0.1±0.2 | 89.3±0.8 | -1.1±0.1 |
| | GCN-Clu | 76.4±0.1 | -14.9±0.3 | 91.1±0.3 | -0.8±0.2 | 98.0±0.1 | -0.6±0.1 | 81.1±0.4 | -3.2±0.6 |
| | GCN-Par | 86.7±0.1 | -7.2±0.1 | 94.6±0.3 | -0.8±0.4 | 98.9±0.3 | -0.4±0.2 | 86.0±0.3 | -6.4±0.2 |
| | GCN-Comp | 86.7±0.3 | -4.9±0.3 | 82.3±1.2 | -2.7±0.6 | 96.8±0.4 | 1.8±0.4 | 79.2±0.4 | 1.2±0.7 |
| | RieGrace | 75.5±0.6 | 0.1±0.2 | 75.4±0.1 | -0.2±0.3 | 67.9±0.1 | -1.1±0.1 | 73.1±0.3 | 0.0±0.0 |
| | TRACE | 94.4±0.1 | 0.4±0.2 | 93.2±0.3 | **3.1±0.7** | 99.5±0.1 | 0.1±0.1 | 88.1±0.2 | 0.1±0.1 |
| | **SAOT** | **96.6±0.3** | **1.5±0.1** | **95.4±0.1** | -0.1±0.1 | **99.5±0.1** | **0.1±0.1** | **95.9±0.3** | **0.2±0.3** |

supervised CGL baselines on four datasets. Specifically, SAOT achieves an improvement of over 15% in terms of AP on the large-scale Products-CL dataset, where severe class imbalance and dense relational structure make cross-task structural preservation essential. SAOT improves the performance by 5% in terms of AP and achieves near-zero forgetting on CoraFull-CL. Furthermore, SAOT attains a score of 98.5% in terms of AP with minimal forgetting on Reddit-CL. Although both CoraFull-CL and Arxiv-CL are citation networks, the stronger semantic overlap and structural homophily in Arxiv-CL result in more severe forgetting under continual learning. The performance matrices in Figure 2 further demonstrate that SAOT sustains more stable accuracy over time and reduces performance degradation across most datasets. Notably, SAOT achieves competi-

tive performance across multiple datasets and has a lower forgetting rate than most supervised learning baselines.

**Task-IL Scenario.** In the Task-IL setting, where task identities are provided during inference, the performance differences among methods typically narrow. The comparison results under the Task-IL scenario are shown in Table 3. We can observe that SAOT outperforms self-supervised baselines across four datasets. Although SAOT performs slightly worse than the TRACE method in AF on the Arxiv-CL dataset, it significantly outperforms TRACE in terms of AP. In continual learning scenarios, a desirable objective is to achieve high overall accuracy while preserving performance on previously learned tasks, which is reflected by high AP and positive AF values, as demonstrated by SAOT.

*Table 4.* Ablation study of SAOT on CoraFull-CL under both Class-IL and Task-IL settings.

| Variants | Class-IL | | Task-IL | |
|---|---|---|---|---|
| | AP (%) | AF (%) | AP (%) | AF (%) |
| w/ Cosine | 68.5 | -3.2 | 96.1 | 1.3 |
| w/ MSE | 69.1 | -3.5 | 96.3 | 1.3 |
| w/o Optimal Transport | 71.6 | -3.2 | 95.2 | 0.8 |
| w/o Cross-task Distill. | 72.3 | -2.2 | 96.2 | 1.1 |
| **SAOT (ours)** | **76.3** | **-0.2** | **96.7** | **1.5** |

### 5.5. Ablation Study

We compare SAOT with its variants to validate the contribution of two components. The designed variants are as follows: (1) To verify the importance of cross-task structural knowledge distillation, we design a variant that removes the distillation, denoted by w/o Cross-task Distillation, where only intra-task graph transport alignment is retained. In addition, to investigate the effect of plan-level distillation compared to point-wise constraints, we design two variants that replace the transport plan distillation with embedding-level objectives based on cosine similarity (w/ Cosine) and mean squared error (w/ MSE), respectively. (2) To examine the importance of optimal transport alignment, we design a variant referred to as w/o Optimal Transport. Specifically, the variant preserves the same replay mechanism as SAOT but replaces the OT-based structural alignment objective with the Graph Barlow Twins objective (Bielak et al., 2022) for representation learning.

The ablation results in Table 4 highlight two key findings: (1) Both cross-task transport-plan distillation and intra-task OT alignment are essential, as removing either component leads to clear drops in AP and more negative AF, especially under the challenging Class-IL setting. (2) Distilling relational structure at the transport-plan level is more effective than point-wise feature constraints, where replacing plan-level distillation with cosine or MSE objectives consistently degrades performance.

### 5.6. Parameter Sensitivity Analysis

We investigate the sensitivity of hyperparameters $\alpha$ and $\beta$, as shown in Figure 3. Firstly, we fix the value of parameter $\beta$ and analyze the effect of parameter $\alpha$ on performance. We can see that as the parameter $\alpha$ increases, SAOT shows a clear improvement in terms of AP on CoraFull-CL. In contrast, SAOT remains relatively insensitive to $\alpha$ on Reddit-CL and achieves the best performance when $\alpha = 0$. These experimental results indicate limited benefits from intra-task structural calibration on large graphs.

We further evaluate $\beta$ with the optimal $\alpha$ on two datasets. SAOT achieves the best trade-off between AP and AF at

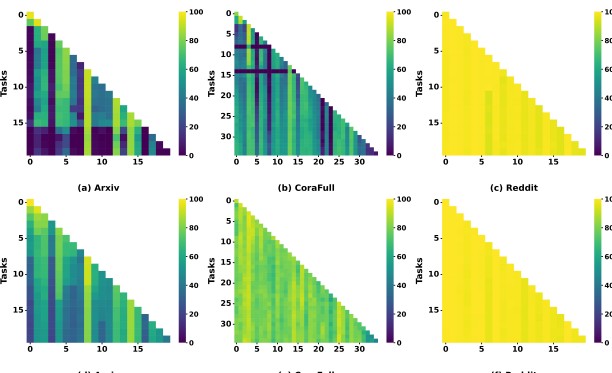

*Figure 2.* Performance matrices of TRACE (top) and SAOT (bottom) on Arxiv-CL, CoraFull-CL, and Reddit-CL under the Class-IL setting.

$\beta = 0.6$ and $\beta = 0.5$ on CoraFull-CL and Reddit-CL, respectively. These results indicate that excessively strong cross-task regularization restricts the plasticity of learning new tasks, which in turn negatively affects stability. Accordingly, we adopt $\beta = 0.6$ for CoraFull-CL and $\beta = 0.5$ for Reddit-CL.

### 5.7. Complexity Analysis

The computational complexity of SAOT mainly comes from graph encoding and the structure-aware optimal transport alignment module. We analyze the complexity on a single task $T_t$ with graph $\mathcal{G}_t = (\mathcal{V}_t, \mathcal{E}_t)$. The graph encoder requires $\mathcal{O}(|\mathcal{V}_t|H^2 + |\mathcal{E}_t|H)$, where $H$ is the embedding dimension. For the structure-aware optimal transport alignment, computing the pairwise transport cost over the full graph requires $\mathcal{O}(|\mathcal{V}_t|^2H)$, while the Sinkhorn optimization introduces an additional $\mathcal{O}(K|\mathcal{V}_t|^2)$ complexity, where $K$ denotes the number of Sinkhorn iterations. As a result, the transport alignment module has quadratic time and memory complexity with respect to the number of graph nodes, making full-graph alignment inefficient for large-scale datasets such as Reddit-CL and Products-CL. To improve scalability, SAOT adopts a point-cloud sampling mechanism that restricts the number of aligned nodes to a fixed budget $M$. The practical complexity of the transport alignment module is reduced to $\mathcal{O}(M^2H + KM^2)$, where $M \ll |\mathcal{V}_t|$ in practice. Figure 4 shows the relation between Class-IL performance of various self-supervised methods and the average running time per task. SAOT achieves a favorable balance between effectiveness and efficiency.

## 6. Conclusion

In this paper, we propose SAOT, a novel self-supervised framework for CGL that leverages the optimal transport mechanism to capture and preserve relational structure among nodes. SAOT constructs optimal transport plans

specifically highlighted here.

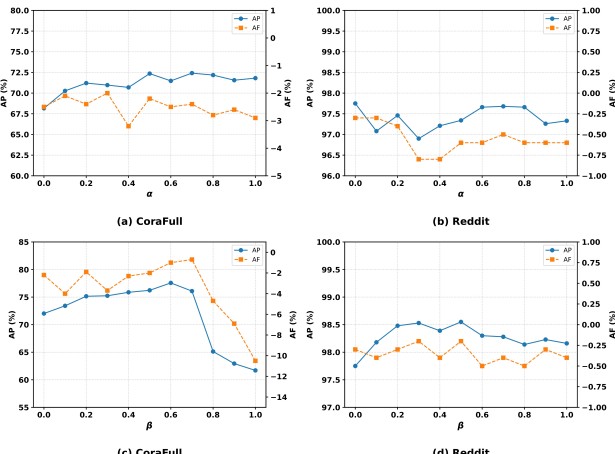

*Figure 3.* Analysis of hyperparameters $\alpha$ and $\beta$ on CoraFull-CL and Reddit-CL datasets under Class-IL setting.

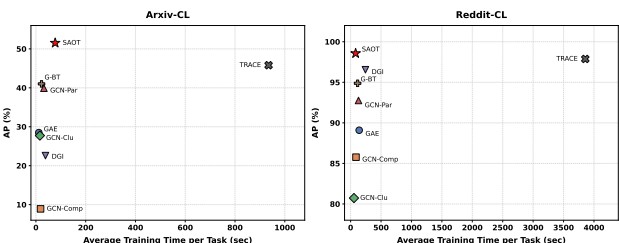

*Figure 4.* The trade-off between AP scores and average running time per task on Arxiv-CL and Reddit-CL datasets under Class-IL setting.

as structure-level references, guiding the encoder to learn high-quality and structure-aware representations. Simultaneously, we incorporate cross-task knowledge distillation to mitigate structural drift caused by continual learning. Extensive experiments show that SAOT consistently outperforms existing self-supervised baselines in terms of AP under both Task-IL and Class-IL settings, and achieves competitive performance compared to supervised methods.

## Acknowledgements

This work is supported in part by National Natural Science Foundation of China (No. 62371340, 62322203), Tianjin Natural Science Foundation Project (No. 24JCZDJC00820, 23JCYBJC00520), Tianjin Measurement Science and Technology Project (NO.2024TJMT061).

## Impact Statement

This work provides a novel self-supervised learning approach for continual graph learning in real-world scenario, and it aims to promote the development of continual learning in machine learning. There are many potential societal consequences of our work, none which we feel must be

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

# A. Datasets Details

The four benchmark datasets for continual graph learning employed in our experiments are derived from four public data sources: OGB-Arxiv, OGB-Products, Reddit, and CoraFull.

*CoraFull:* CoraFull is a citation network dataset extended from the original Cora dataset, covering a broader range of computer science literature. In this article, the dataset consists of 19,793 papers ($N$), 126,842 citations ($E$), and 70 class categories ($C$) based on research fields. The attribute information of papers is extracted as 8,710-dimensional sparse Bag-of-Words (BoW) vectors ($D$). We divide the 70 classes into 35 incremental tasks ($T$) for the continual learning experiments, with each task containing 2 classes.

*OGB-Arxiv:* OGB-Arxiv is a directed citation network constructed from the Microsoft Academic Graph (MAG), representing Computer Science papers. This dataset consists of 169,343 papers ($N$) and 1,166,243 edges ($E$). The nodes are labeled into 40 subject areas ($C$) (e.g., cs.AI, cs.LG). The attribute information is represented by 128-dimensional feature vectors ($D$) obtained by averaging the Word2Vec embeddings of titles and abstracts. We classify the papers into 20 tasks ($T$) according to their subject areas, where each task consists of 2 classes.

*Reddit:* Reddit is a social interaction graph derived from Reddit posts, where nodes represent posts and edges represent user co-interaction. A subset of Reddit which contains 227,853 posts ($N$) and approximately 114.6 million edges ($E$) is extracted after filtering. The original dataset contains 41 communities; we remove one class with insufficient samples to retain 40 classes ($C$). The attribute information of posts is extracted as 602-dimensional vectors ($D$) from GloVe word embeddings and metadata. We divide the communities into 20 tasks ($T$) for use with experiments.

*OGB-Products:* OGB-Products is an undirected co-purchasing network representing products sold on Amazon. This dataset consists of 2,449,028 products ($N$) and 61,859,036 edges ($E$). The products are categorized into 47 classes; we exclude one class (containing only a single node) to keep 46 classes ($C$). The attribute information is 100-dimensional ($D$), generated by applying PCA to bag-of-words features from product descriptions. We divide the products into 23 tasks ($T$), with each task containing 2 classes.

**Data Resources:**

The datasets used in this study are publicly available from the following sources:

**OGB-Arxiv:** Available from https://ogb.stanford.edu/docs/nodeprop/#ogbn-arxiv.

**OGB-Products:** Available from https://ogb.stanford.edu/docs/nodeprop/#ogbn-products.

**Reddit:** Available from https://archive.org/details/FullRedditSubmissionCorpus2006ThruAugust2015.

# B. Sensitivity Analysis of Replay Buffer Size

To evaluate the dependence of SAOT on the memory buffer capacity, we conduct a sensitivity analysis by varying the replay buffer size $|\mathcal{B}|$ across all four datasets under both Class-IL and Task-IL settings. We test buffer sizes ranging from 0 (no replay) up to 4,000 samples (depending on the dataset scale), measuring both Average Performance (AP) and Average Forgetting (AF). The results are illustrated in Figure 4 (Class-IL) and Figure 5 (Task-IL).

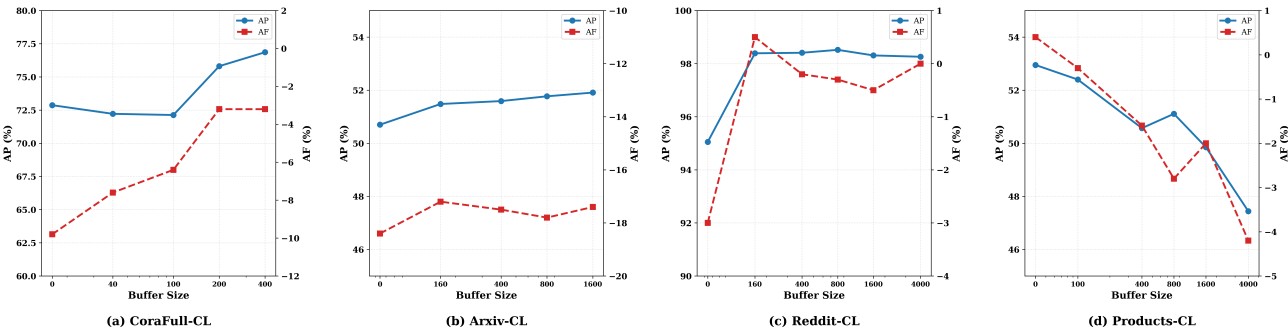

*Figure 5.* Sensitivity analysis of buffer size under the Class-IL setting.

*Figure 6.* Sensitivity analysis of buffer size under the Task-IL setting.

**Low Dependency on Explicit Memory:** In most scenarios, SAOT achieves near-optimal or optimal performance with zero or minimal replay budgets (e.g., buffer sizes of 0 or 160). Especially in the Task-IL setting, the performance curves remain remarkably flat regardless of the buffer size. This indicates that the structural knowledge acquired via optimal transport alignment is intrinsically robust, mitigating forgetting without relying on the rehearsal of historical samples.

**Effectiveness of Structural Alignment over Replay:** In the Class-IL setting, increasing the buffer size does not consistently yield performance gains; for instance, on Products-CL, performance even fluctuates as the buffer size increases. This observation reinforces that the effectiveness of SAOT stems from its structure-aware design rather than rote memorization. The framework is capable of capturing and preserving essential topological structures with minimal resources, whereas excessive replay might introduce structural noise or inductive bias from the buffered subgraphs.

