# OpenReview forum: "SAOT: Self-Supervised Continual Graph Learning with Structure-Aware Optimal Transport"
_ICML.cc/2026/Conference — ICML 2026 regular_

### Official Review · Reviewer_DDmC · 2026-03-06

**Soundness:** 3
**Presentation:** 3
**Significance:** 3
**Originality:** 3
**Overall Recommendation:** 5
**Confidence:** 4

**Summary:**

This paper proposes SAOT, a novel self-supervised continual graph learning method that explicitly captures and preserves relational structure within graph representations across sequential tasks. Simultaneously, SAOT incorporates a cross-task knowledge distillation mechanism to preserve the previous structural knowledge. Extensive experiments conducted on multiple datasets demonstrate the state-of-the-art performance of the proposed SAOT.

**Compliance With Llm Reviewing Policy:**

Affirmed.

**Final Justification:**

All of my concerns have been satisfactorily addressed. Overall, this paper is intereting and well-motivated, I maintain my positive score.

**Key Questions For Authors:**

(1) While SAOT achieves the best AP on Arxiv-CL under Class-IL, its AF is significantly worse than TRACE (-6.4%) and even some supervised methods. A deeper analysis of this specific failure case would be insightful.

(2) Some equations (e.g., Equations 5) need more intuitive explanations.

**Limitations:**

yes

**Strengths And Weaknesses:**

Strengths:
(1) The paper identifies a significant and nuanced problem in self-supervised continual graph learning (CGL)—structural drift—which is often overlooked by instance-level methods. The proposed solution, using Optimal Transport to capture and preserve relational structures, is a novel and theoretically grounded approach to this problem.
(2) The SAOT framework is well-structured and clearly explained.
(3) Experimental results demonstrate that SAOT achieves significant performance gains in the most of the datasets.

Weaknesses:
(1) A major concern for OT-based methods, especially those involving Gromov-Wasserstein distances (Equation 5-7), is their computational cost. The paper mentions using a "batched" approach but does not provide a detailed analysis of the time and memory complexity of computing the fused optimal transport plans.
(2)The notation G_t^bat is introduced in Section 4 but its precise meaning is not fully clear.

---

> ### Author Rebuttal · Authors · 2026-03-31
>
> Strengths And Weaknesses:
>
> Q1: A major concern for OT-based methods, especially those involving Gromov-Wasserstein distances (Equation 5-7), is their computational cost. The paper mentions using a "batched" approach but does not provide a detailed analysis of the time and memory complexity of computing the fused optimal transport plans.
>
> R1: Many thanks. For the graph with $N$ nodes, the complexity of OT distance is $\mathcal{O}(N^2)$. To address this computational bottleneck and ensure scalability on large-scale graphs, our method is designed by a point-cloud sampling mechanism. By restricting the maximum number of aligned points to $M$, the time complexity is strictly bounded to $\mathcal{O}(K \cdot M^2)$, where $K$ is the number of Sinkhorn iterations. We will add this analysis to the revised manuscript. The total running time (in seconds) compared to baselines are shown as follows.
> | **Method** | **Arxiv-CL** | **Reddit-CL** |
> | ---------- | ------------ | ------------- |
> | GAE        | 231.87s       | 2810.49s       |
> | DGI        | 767.05s       | 4865.15s       |
> | G-BT       | 450.73s       | 2294.32s       |
> | GCN-Clu    | 318.55s       | 1099.78s       |
> | GCN-Par    | 643.61s       | 2590.60s       |
> | GCN-Comp   | 372.95s       | 1762.04s       |
> | TRACE      | 16064.00s     | 77149.03s      |
> | SAOT       | 1548.00s     | 1936.09s       |
>
>
> Q2: The notation G_t^bat is introduced in Section 4 but its precise meaning is not fully clear.
> R2: Thank you for the comment. $\mathcal{G}_t^{\textit{bat}}$ denotes the batched graph formed by merging the current task graph $\mathcal{G}_t$ with historical subgraphs from the memory buffer via a disjoint union. This combined graph serves as the training input for task $\mathcal{T}_t$. We will clarify this in the revised manuscript.
>
> Key Questions For Authors:
>
> Q1: While SAOT achieves the best AP on Arxiv-CL under Class-IL, its AF is significantly worse than TRACE (-6.4%) and even some supervised methods. A deeper analysis of this specific failure case would be insightful.
>
> R1: Thank for your comments. We note that Arxiv-CL under the Class-IL setting is generally considered more challenging, where strong forgetting has been observed across many methods. This suggests that the distribution discrepancies across tasks in Arxiv-CL are relatively large, making it more difficult to simultaneously retain previously learned knowledge and adapt to new classes. SAOT leverages optimal transport to model and preserve relational structure in graph representations across sequential tasks. When task distributions differ substantially, preserving relational structure alone may be insufficient to fully retain previously learned knowledge, which can lead to relatively higher forgetting.
>
> Q2: Some equations (e.g., Equations 5) need more intuitive explanations.
>
> R2: Many thanks. Equation 5 is the fused Gromov-Wasserstein distance (considering both nodes and edges) commonly used for OT. Equation 6 is a shorthand for the right side of the equal sign in Equation 5. Equations 5 and 6 are used to calculate the optimal transport plan  from the corresponding OT distances. We will provide a more detailed explanation of the formula in the final version.

---

> > ### Author Rebuttal · Reviewer_DDmC · 2026-04-01
> >
> > I appreciate the authors' thorough responses to my questions. All of my concerns have been satisfactorily addressed.  Overall, this paper is intereting and well-motivated,  I maintain my positive score.

---

### Official Review · Reviewer_ip6V · 2026-03-08

**Soundness:** 3
**Presentation:** 3
**Significance:** 2
**Originality:** 2
**Overall Recommendation:** 5
**Confidence:** 2

**Summary:**

Summary:

The author proposes a new method called SAOT, a self-supervised continual graph learing framework which uses structure-aware optimal transport to preserve global relational structure. SAOT has achieved SOTA performance on 4 benchmark datasets compared with baselines.

**Compliance With Llm Reviewing Policy:**

Affirmed.

**Final Justification:**

Thanks for the rebuttal. My concerns have been mostly addressed and thus raise the score to 5.

**Key Questions For Authors:**

please refer to cons

**Limitations:**

Yes

**Strengths And Weaknesses:**

Pros:
- The author proposes a new method called SAOT, a self-supervised continual graph learing framework which uses structure-aware optimal transport to preserve global relational structure.

- SAOT has achieved SOTA performance on 4 benchmark datasets compared with baselines.

Cons:
- The experiments are conducted on only 4 datasets, which seems limited.
- More extreme setting, like replay-free would be interesting to explore and test
- OT is computationally heavy and the efficiency analysis is not comprehensive.

---

> ### Author Rebuttal · Authors · 2026-03-31
>
> Strengths And Weaknesses:
>
> Q1: The experiments are conducted on only 4 datasets, which seems limited.
>
> R1: Many thanks. The four datasets used in our experiments are commonly used benchmark datasets in the field of Continual Graph Learning and have been widely adopted in evaluating related methods, as referenced in the paper CGLB: Benchmark Tasks for Continual Graph Learning, NeurIPS 2022. We acknowledge that evaluating on additional datasets would be beneficial and will explore it in future work.
>
> Q2: More extreme setting, like replay-free would be interesting to explore and test.
>
> R2: Thank for your comments. We have tested SAOT under a replay-free setting in the Appendix B, as shown in Figure 4 and Figure 5. While the model performs well in many scenarios, we observe that under a replay-free setting, particularly on datasets like CoraFull-CL and Reddit-CL, the performance is not as strong compared to scenarios with replay. This indicates that, while SAOT is designed to handle structure-aware learning, the absence of explicit memory replay does impact its ability to preserve past knowledge in certain tasks. We will discuss these findings and explore potential improvements for replay-free scenarios in the future work section of the manuscript.
>
> Q3: OT is computationally heavy and the efficiency analysis is not comprehensive.
>
> R3: Thank for your comment. For the graph with $N$ nodes, the complexity of OT distance with Sinkhorn algorithm is $\mathcal{O}(N^2)$. To address this computational bottleneck and ensure scalability on large-scale graphs, our method is designed by a point-cloud sampling mechanism. By restricting the maximum number of aligned points to $M$, the time complexity is strictly bounded to $\mathcal{O}(K \cdot M^2)$, where $K$ is the number of Sinkhorn iterations. We will add this analysis to the revised manuscript. The total running time (in seconds) compared to baselines are shown as follows.
> | **Method** | **Arxiv-CL** | **Reddit-CL** |
> | ---------- | ------------ | ------------- |
> | GAE        | 231.87s       | 2810.49s       |
> | DGI        | 767.05s       | 4865.15s       |
> | G-BT       | 450.73s       | 2294.32s       |
> | GCN-Clu    | 318.55s       | 1099.78s       |
> | GCN-Par    | 643.61s       | 2590.60s       |
> | GCN-Comp   | 372.95s       | 1762.04s       |
> | TRACE      | 16064.00s     | 77149.03s      |
> | SAOT       | 1548.00s     | 1936.09s       |

---

> > ### Author Rebuttal · Reviewer_ip6V · 2026-04-03
> >
> > Thanks for the rebuttal and response to my concern. My concerns have been mostly addressed

---

### Official Review · Reviewer_8vrM · 2026-03-09

**Soundness:** 3
**Presentation:** 3
**Significance:** 2
**Originality:** 2
**Overall Recommendation:** 3
**Confidence:** 4

**Summary:**

This work identifies that current self-supervised CGL methods often rely on instance-level consistency, which can lead to structural drift where the global relationships between node embeddings distort over time. The proposed SAOT method addresses this problem by using Optimal Transport theory to capture and align the global relational structure between the input graph space and the representation space. Experiments are conducted on 4 benchmark datasets to demonstrate the benefit of the proposed approach.

**Compliance With Llm Reviewing Policy:**

Affirmed.

**Final Justification:**

The authors addressed most of my concerns, although i feel the novelty is still relatively insufficient. I'm raising my score from 2 to 3.

**Key Questions For Authors:**

Please see weaknesses above.

**Limitations:**

Not much discussion. One limitation authors could potentially discuss: there are different kinds of evolving distributions (e.g. gradual drift or a burst/rapid changes). Can the proposed method work equally work well in these different distributions, or perform better for certain scenarios?

**Strengths And Weaknesses:**

Strengths:

1. SAOT explicitly models inter-node correspondences, effectively mitigating structural drift.

2. SAOT seems to be quite effective on the selected datasets.



Weaknesses:

1. The manuscript lacks a rigorous theoretical analysis to demonstrate the advantages of SAOT. Given the high standards for theoretical depth at ICML, the paper may not be suitable in its current form.

2. I have concerns about the novelty. The application of OT to the graph domain is already a well-established paradigm.  For example, the following papers also apply the optimal transport theory. What are the key distinction in this paper?

 [1] HGOT: Self-supervised Heterogeneous Graph Neural Network with Optimal Transport. ICML 2025.

 [2] GCL-OT: Graph Contrastive Learning with Optimal Transport for Heterophilic Text-Attributed Graphs. AAAI 2026.

3. Following 2, calculating OT distance  can be computationally intensive for very large graphs compared to simple point-wise CGL objectives. Therefore, both complexity and running time compared to baselines should be reported.

4. “The learning rate is tuned within the range 1e-4 to 1e-2”. A more detailed setting that has been adopted should be reported.

---

> ### Author Rebuttal · Authors · 2026-03-31
>
> Q1: The manuscript lacks a rigorous theoretical analysis to demonstrate the advantages of SAOT. Given the high standards for theoretical depth at ICML, the paper may not be suitable in its current form.
>
> R1: Many thanks. For continual graph learning, long-term knowledge in graph data is encoded not only in individual node embeddings but also in relational structures. Let $\pi\_{\mathcal{G}}^{\*}$ and $\pi\_{\mathcal{Z}}^{\*}$ be the optimal transport plans in graph and representation spaces defined in Eq. (7) and Eq. (8). Since $\pi\_{\mathcal{G}}^{\*}$ is obtained by minimizing the FGW objective defined on node features and graph topology, it captures structure-aware correspondences in the input graph. Minimizing the alignment loss: $$\mathcal{L}\_{mat} = \Theta(\pi\_{\mathcal{G}}^{\*}, \pi\_{\mathcal{Z}}^{\*})$$
> force $\pi\_{\mathcal{Z}}^{\*}$ to match these correspondences, ensuring that structurally related nodes remain relationally consistent in the embedding space.
> Furthermore, let $\pi_Z^{t-1,}$ and $\pi_Z^{t,}$ denote the transport plans from the teacher and current models on the same input graphs. Using triangle inequality of the OT divergence:
> $$\Theta(\pi\_{\mathcal{Z}}^{t-1,\*}, \pi\_{\mathcal{Z}}^{t,\*}) \leq \Theta(\pi\_{\mathcal{Z}}^{t-1,\*}, \pi\_{\mathcal{G}}^{\*}) + \Theta(\pi\_{\mathcal{G}}^{\*}, \pi\_{\mathcal{Z}}^{t,\*})$$
> The left side is exactly the cross-task distillation objective $\mathcal{L}\_{skd}$. The right side bounds this drift via a constant (the frozen teacher's alignment loss) and the current intra-task alignment loss $\mathcal{L}\_{mat}$.
> Thus, jointly minimizing both objectives theoretically bounds structural drift between tasks, explaining how SAOT captures and preserves relational structure. Due to space limitations, the theoretical discussion was not included in the manuscript, and we will add it in the appendix in the revised version.
>
> Q2: I have concerns about the novelty. The application of OT to the graph domain is already a well-established paradigm. For example, the following papers also apply the optimal transport theory. What are the key distinction in this paper?
>
> R2: Thanks. The SAOT has two key advantages: (1) it is the first to leverage Optimal Transport for preserving global inter-node relationships across tasks in continual graph learning; (2) it introduces cross-task knowledge distillation to mitigate structural drift in dynamic graphs. These contributions distinguish SAOT from existing methods such as HGOT and GCL-OT, which primarily focus on static graph representations.
>
> Q3: Following 2, calculating OT distance can be computationally intensive for very large graphs compared to simple point-wise CGL objectives. Therefore, both complexity and running time compared to baselines should be reported.
>
> R3: Thank you for your comment. For the graph with $N$ nodes, the complexity of OT distance is $\mathcal{O}(N^2)$. To address this computational bottleneck and ensure scalability on large-scale graphs, our method is designed by a point-cloud sampling mechanism. By restricting the maximum number of aligned points to $M$, the time complexity is strictly bounded to $\mathcal{O}(K \cdot M^2)$, where $K$ is the number of Sinkhorn iterations. The total running time (in seconds) compared to baselines are shown as follows.
> | **Method** | **Arxiv-CL** | **Reddit-CL** |
> | ---------- | ------------ | ------------- |
> | GAE        | 231.87s       | 2810.49s       |
> | DGI        | 767.05s       | 4865.15s       |
> | G-BT       | 450.73s       | 2294.32s       |
> | GCN-Clu    | 318.55s       | 1099.78s       |
> | GCN-Par    | 643.61s       | 2590.60s       |
> | GCN-Comp   | 372.95s       | 1762.04s       |
> | TRACE      | 16064.00s     | 77149.03s      |
> | SAOT       | 1548.00s     | 1936.09s       |
>
> Q4: “The learning rate is tuned within the range 1e-4 to 1e-2”. A more detailed setting that has been adopted should be reported.
>
> R4: Thank you. In our experiments, the learning rate is determined by using the grid-search strategy. For the Arxiv-CL, CoraFull-CL, Reddit-CL, and Products-CL datasets, the learning rates are set as 0.001, 0.01, 0.001, and 0.01, respectively.
>
> Q5: Not much discussion. One limitation authors could potentially discuss: there are different kinds of evolving distributions (e.g. gradual drift or a burst/rapid changes). Can the proposed method work equally work well in these different distributions, or perform better for certain scenarios?
>
> R5: Many thanks. We follow the task construction in CGLB and TRACE, where tasks are created by splitting data into disjoint subsets, so our SAOT performs well for the rapid scenario. Gradual drift can be viewed as a sequence of small shifts between consecutive tasks. Since SAOT aligns relational structures via optimal transport, the same mechanism can also work under gradual drift. It is worth noting that SAOT does not specifically account for very long-range, gradual distribution drifts across many small steps.

---

> > ### Author Rebuttal · Reviewer_8vrM · 2026-04-04
> >
> > The authors addressed most of my concerns, although the novelty aspect may still be a concern. However, I can raise my score from 2 to 3, and I will not object to acceptance if there are other reviewers championing the acceptance. Thank you.

---

> > > ### Author Response · Authors · 2026-04-08
> > >
> > > Q: The authors addressed most of my concerns, although the novelty aspect may still be a concern.
> > >
> > > R: Thanks. Prior works such as HGOT and GCL-OT focus on static graph settings, where optimal transport is employed to match different views (e.g., structure–text or multi‑meta‑path views) and thereby improve representation quality within a single fixed graph. In contrast, our work aims to mitigate catastrophic forgetting and structural drift in self-supervised continual graph learning scenarios. In SAOT, the transport plans are treated as structural knowledge carriers and explicitly distilled across tasks, enforcing consistency of inter-node relationships over time. By preserving OT-induced relational structure rather than individual embeddings, SAOT mitigates structural distortion during sequential updates.

---

### Official Review · Reviewer_bGxt · 2026-03-12

**Soundness:** 3
**Presentation:** 3
**Significance:** 3
**Originality:** 4
**Overall Recommendation:** 5
**Confidence:** 4

**Summary:**

This paper addresses a critical and under-explored problem in continual graph learning (CGL): self-supervised learning that preserves global relational graph structure to mitigate structural drift and catastrophic forgetting. By aligning OT plans from the graph space and the representation space, SAOT guides the encoder to learn embeddings that retain global relational structure; by distilling OT plans from previous tasks, it mitigates structural drift in CGL. Extensive experiments conducted on multiple datasets demonstrate its performance.

**Compliance With Llm Reviewing Policy:**

Affirmed.

**Final Justification:**

The rebuttal has addressed my concerns. I maintain my original score.

**Key Questions For Authors:**

See weakness

**Limitations:**

yes

**Strengths And Weaknesses:**

Strengths:

(1)The core contribution of integrating Structure-Aware Optimal Transport (SAOT) into self-supervised CGL is innovative. Unlike existing instance-level consistency methods that optimize isolated node embeddings, it explicitly models global inter-node correspondences via OT theory and Gromov-Wasserstein distance.

(2)The design of a transport plan-level distillation mechanism (rather than point-wise embedding constraints) is a well-justified choice.

(3) The problem formulation for self-supervised CGL is clear and consistent with prior work, and the decoupled representation-classifier paradigm ensures fair evaluation of learned embeddings.

Weaknesses:

(1) The paper provides no quantitative analysis of whether it is sensitive to task order, which is a common issue in continual learning. The experiments use a fixed task order, and no random task order evaluations are included, limiting the generalizability of the results.

(2)The authors use a 2-layer GCN for most datasets and a 3-layer GAT for Reddit-CL but provide no justification for this choice (e.g., why GAT is better for Reddit-CL, or how encoder depth/architecture impacts SAOT’s performance).

---

> ### Author Rebuttal · Authors · 2026-03-31
>
> Strengths And Weaknesses:
>
> Q1: The paper provides no quantitative analysis of whether it is sensitive to task order, which is a common issue in continual learning. The experiments use a fixed task order, and no random task order evaluations are included, limiting the generalizability of the results.
>
> R1: Many thanks. Task order sensitivity is indeed important. Following CGLB and prior work, we used a fixed order for fair comparison. In the revision, we will add experiments with random task orders to further assess SAOT's robustness.
>
> Q2: The authors use a 2-layer GCN for most datasets and a 3-layer GAT for Reddit-CL but provide no justification for this choice (e.g., why GAT is better for Reddit-CL, or how encoder depth/architecture impacts SAOT’s performance).
>
> R2: Thank for your comments. Reddit-CL has substantially higher edge density (over 100M edges for ~227K nodes) compared to the other datasets, resulting in extremely large neighborhoods. In this setting, a 2-layer GCN tends to aggregate excessive neighbor information. Therefore, we adopt a 3-layer GAT for Reddit-CL, as its attention mechanism can selectively focus on relevant neighbors and better handle dense structures. We will clarify this choice in the revised manuscript.
>
> [1] CGLB: Benchmark Tasks for Continual Graph Learning. NeurIPS 2022.

---

> > ### Author Rebuttal · Reviewer_bGxt · 2026-04-03
> >
> > The rebuttal has addressed my concerns. I maintain my original score.

---

### Decision · Program_Chairs · 2026-04-30

**Decision:**

Accept (regular)

**Comment:**

The paper has four reviews, with ratings 5, 5, 5, and 3, respectively. The reviewers raised the following major concerns:
1. The computational cost is high (resolved by the rebuttal).
2. The novelty and theoretical analysis are insufficient (from Reviewer 8vrM).
3. The tested datasets are insufficient (resolved by the rebuttal).

I partially agree with the concern of limited novelty in applying OT to graph learning. However, the application of OT to continual graph learning seems novel. Moreover, the experiments showed that the proposed method is much more effective than the baselines, meaning that the innovation is significant.

During the discussion, Reviewer 8vrM has no obligation to accept the paper.

Based on the above, I recommend accepting the paper.